# ACCELERATING RETRIEVAL-AUGMENTED LANGUAGE MODEL SERVING WITH SPECULATION

## ABSTRACT

Retrieval-augmented language models (RaLM) have demonstrated the potential to solve knowledge-intensive natural language processing (NLP) tasks by combining a non-parametric knowledge base with a parametric language model. Instead of fine-tuning a fully parametric model, RaLM excels at its low-cost adaptation to the latest data and better source attribution mechanisms. Among various RaLM approaches, iterative RaLM delivers a better generation quality due to a more frequent interaction between the retriever and the language model. Despite the benefits, iterative RaLM usually encounters high overheads due to the frequent retrieval step. To this end, we propose RaLMSpec, a speculation-inspired framework that provides generic speed-up over iterative RaLM while preserving the same model outputs through *speculative retrieval* and *batched verification*. By further incorporating prefetching, optimal speculation stride scheduler, and asynchronous verification, RaLMSpec can automatically exploit the acceleration potential to the fullest. Extensive evaluations over three language models on four downstream QA datasets demonstrate that RaLMSpec can achieve a speed-up ratio of 1.75-2.39×, 1.04-1.39×, and 1.31-1.77× when the retriever is an exact dense retriever, approximate dense retriever, and sparse retriever respectively compared with the baseline.

## 1 INTRODUCTION

Recent advancements in large language models such as LLaMA-2, GPT-3, and PaLM have shown promising results in diverse NLP tasks (Touvron et al., 2023; Brown et al., 2020; Chowdhery et al., 2022). However, encoding a massive amount of knowledge into a fully parametric model requires excessive effort in both training and deployment. The situation can be further exacerbated when the foundation model is required to adapt to new data or various downstream tasks (Asai et al., 2023). To address this challenge, recent work introduces *retrieval-augmented* language models (RaLM), which integrate the parametric language model with a non-parametric knowledge base through retrieval augmentation (Guu et al., 2020; Shi et al., 2023; Ram et al., 2023; Khattab et al., 2022).

Existing RaLM methods can be categorized into two classes based on the interaction between the knowledge base and language model. First, *one-shot* RaLM performs retrieval *once* for each request and combines the retrieved documents with the original request to assist generation. On the other hand, *iterative* RaLM enables iterative interactions between the language model and knowledge base for a request so that the language model can opportunistically query the knowledge base to retrieve more relevant documents during generation. Compared to iterative RaLM, one-shot RaLM introduces less retrieval overhead but is inherently limited when the required information varies from time to time during the generation process. On the other hand, iterative RaLM achieves better generative performance while suffering from frequent retrievals and thus high retrieval overhead. This paper answers the following research question: *can we reduce the overhead of iterative RaLM without affecting generative quality?*

We propose RaLMSpec, a framework that employs *speculative retrieval* with *batched verification* to reduce the serving overhead for iterative RaLM while provably preserving the model output. A key bottleneck of existing iterative RaLM methods is the inefficiency of retrieval. In particular, due to the auto-regressive nature of generative language models, the retrieval step is usually performed with a single query summarizing the current context. As shown in Figure 1(a), existing iterative RaLM approaches interleave the retrieval step and the generation step by constantly retrieving from the

knowledge base with the latest context-dependent queries (i.e., $q_0, q_1$, and $q_2$). The corresponding retrieved contents (A, B, C) can then assist the generation process by contributing relevant information to the language model through a prompt or attention-level combination. However, issuing these queries for knowledge base retrieval *sequentially* is intrinsically inefficient.

The idea of speculative retrieval is conceptually similar to speculative execution originated from the computer architecture literature (Burton, 1985). More specifically, RaLMSpec replaces the expensive, iterative retrieval steps of existing RaLM methods with more efficient but less accurate speculative retrieval steps. Consequently, RaLMSpec uses a *batched verification* step to correct any incorrect speculation resul and preserve the model's generative quality. More precisely, after a number of speculative retrieval steps, RaLMSpec initiates a verification step by performing a batched retrieval (i.e., ⑥ in Figure 1(b)), where the queries in the batch are the corresponding queries in the speculative retrieval steps. If there is a mismatch between the speculated documents and the ground truth documents retrieved in the verification step, RaLMSpec automatically corrects the mismatch by rolling back to the first mis-speculated position and rerunning language model decoding using the ground truth documents. As a result, a subpar speculation method with an early mismatch can result in additional overhead. In observing that the same entry can be repetitively retrieved from the knowledge base when generating the response (i.e., temporal locality), RaLMSpec maintains a *local cache* to store past documents for each request, and performs speculative retrieval by retrieving from the local cache instead of the knowledge base. RaLMSpec updates the local cache by directly adding documents retrieved from the knowledge base in each verification step. Figure 1(c) shows a timeline comparison between RaLMSpec and existing iterative RaLM methods. RaLMSpec's latency saving is essentially obtained through efficient batched retrieval, i.e. retrieving from the knowledge base with $n$ queries is more efficient than executing $n$ retrievals sequentially. We show evidence of the above claim in Appendix A.1.

Besides maintaining a local cache for speculative retrieval, we further propose three additional techniques to further reduce RaLM serving latency. First, RaLMSpec can support *cache prefetching* by updating the local cache with the top-k retrieved documents from the knowledge base to boost RaLMSpec's speculative performance. Second, RaLMSpec enables an *optimal speculation stride scheduler* that dynamically adjusts the speculation stride (i.e., the number of consecutive speculation steps between two verification steps) to minimize the speculation overhead. Third, RaLMSpec can exploit concurrency by allowing *asynchronous verification*, which enables an extra speculation step to be performed asynchronously with a verification step.

Extensive evaluation of RaLMSpec over the Wiki-QA (Yang et al., 2015), Web Questions (Berant et al., 2013), Natural Questions (Kwiatkowski et al., 2019), and Trivia QA (Joshi et al., 2017) datasets on the GPT-2, OPT, and LLaMA-2 models shows that RaLMSpec can automatically adapt the speculation configuration and reduce the serving latency of the baseline implementation by up to $2.4\times$, $1.4\times$, $1.8\times$ with an exact dense, approximate dense, and sparse retriever, respectively.

**Contributions.** This paper makes the following contributions:

- We propose RaLMSpec, a framework that reduces the serving latency of generic iterative RaLM approaches while preserving the same model outputs.
- Technically, by leveraging the temporal locality of the retrieved documents, RaLMSpec uses a caching-based speculative retrieval mechanism with batched verification to reduce the retrieval overhead. We further propose three additional techniques to reduce RaLM serving latency, namely cache prefetching, asynchronous verification, and optimal speculation stride scheduling.
- Empirically, we validate that RaLMSpec achieves significant RaLM serving latency reduction across different datasets, language models, and retriever types. These results indicate RaLMSpec can be a generic acceleration framework for serving iterative RaLMs.

## 2 RELATED WORK

**Retrieval-augmented language models.** Since Guu et al. (2020) first proposes to provide relevant information to the language model with retrieved documents from an external knowledge base, numerous works have started to leverage retrieval to improve the language model generation quality (Shi et al., 2023; Park et al., 2023; Wang et al., 2023a; Zhu et al., 2023; Rubin & Berant, 2023;

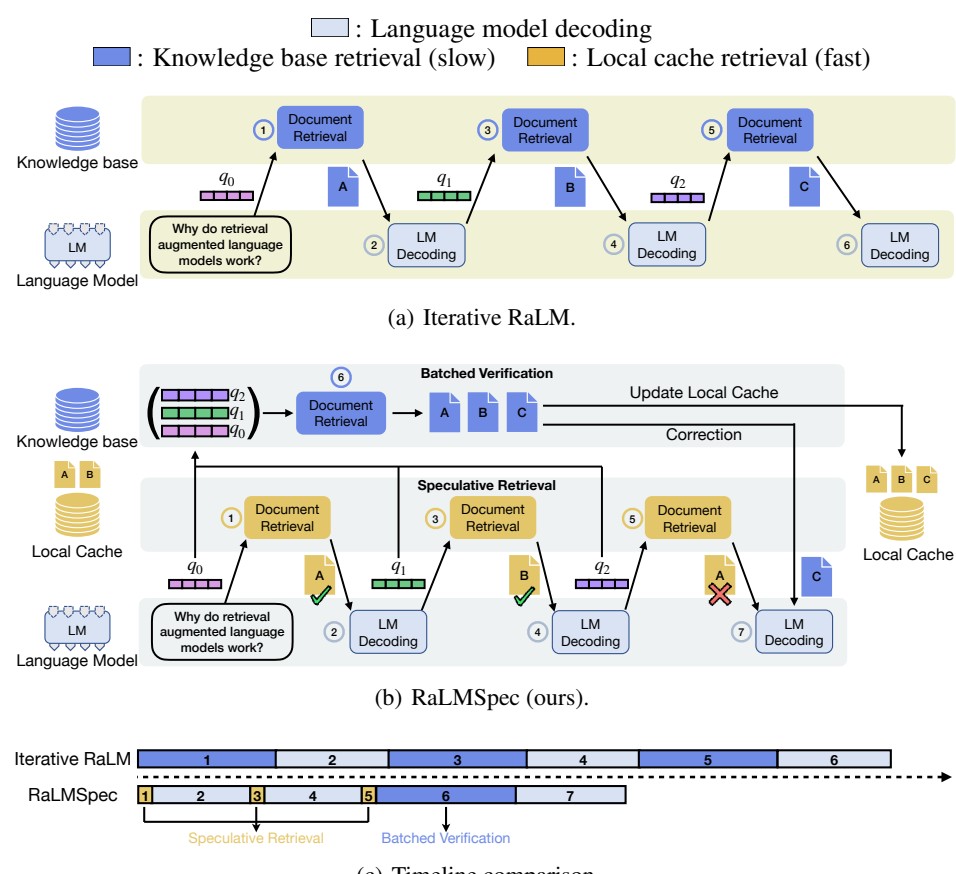

Figure 1: $\{q_0, q_1, q_2\}$ denotes context-dependent query embeddings and A, B, C are document entries. Figure 1(a) shows the workflow of existing iterative RaLM, which suffers from high retrieval overhead. Figure 1(b) shows an overview of RaLMSpec, which enables faster speculative retrieval steps (①, ③, ⑤) followed by a batched verification step (⑥) to guarantee correctness. Consequently, RaLMSpec achieves a lower latency while preserving model quality as shown in Figure 1(c).

Wang et al., 2023b; Zhou et al., 2023). As these works only perform retrieval once before the language model generation starts, we refer to them as one-shot RaLM. Besides one-shot RaLM approaches, another line of work performs retrieval regularly when serving a single request. Ram et al. (2023); Lewis et al. (2020); Jiang et al. (2023); Borgeaud et al. (2022); Khattab et al. (2022) retrieve constantly from the external database with the latest context and leverage the retrieved information to improve the generation quality either through direct concatenation or intermediate layer cross-attention. K-Nearest Neighbour Language Models (KNN-LM), on the other hand, produces the next token distribution by interpolating between a weighted distribution over $k$ retrieved documents and the language model output (Khandelwal et al., 2019; Drozdov et al., 2022). Compared with one-shot RaLM, iterative RaLM methods have been shown to provide higher quality responses at the cost of excessive latency overhead (Khandelwal et al., 2019; Drozdov et al., 2022; Ram et al., 2023). For instance, Khandelwal et al. (2019) retrieves up to 1024 documents for each token generated, which results in unaffordable serving overhead in practice. Reducing the serving overhead of iterative RaLM methods while preserving its high-quality output is thus the core of our work.

**Retrievers for RaLM.** Different retrievers have different tradeoffs between retrieval overheads and retrieval accuracy when serving RaLMs. Instead of using conventional sparse retrievers such as TF-IDF or BM25 (Ramos et al., 2003; Robertson et al., 2009), Karpukhin et al. (2020) trains a dense retriever particularly for RaLM. Similarly, Izacard et al. (2021) proposes to train a dense retriever by unsupervised contrastive learning. Later work further explores the possibility of end-to-end pretraining, where the retriever and language model are trained collaboratively (Izacard et al., 2022; Zhong et al., 2022; Khattab & Zaharia, 2020; Santhanam et al., 2021). Exact dense retrievers are inefficient but accurate, while approximate retrievers are fast to query but less accurate (He et al.,

2021; Xiong et al., 2020). To demonstrate the generality of our design, we experiment RaLMSpec with different retrievers (sparse, exact dense, and approximate dense retrievers).

**Iterative RaLM serving.** As for efficient iterative RaLM serving approaches, the work most relevant to ours is Alon et al. (2022). By using a pre-computed automaton state when a complete retrieval for the KNN-LM is unnecessary, Alon et al. (2022) can reduce the number of calls to the external knowledge base and thus save latency. However, Alon et al. (2022) is not guaranteed to preserve the same model output and hence might compromise model generation quality. To this end, our goal is to achieve generic and efficient serving for existing iterative RaLM approaches while guaranteeing to preserve model quality. To the best of our knowledge, RaLMSpec is the first work that achieves inference time speed-up for generic iterative RaLM approaches without compromising model output. The key intuition behind RaLMSpec is speculative retrieval and batched verification. Speculation has a long history in the computer architecture field (Burton, 1985). Recent works further bring the concept of speculative decoding into Large Language Models (LLM) serving, which essentially reduces serving latency (Leviathan et al., 2022; Stern et al., 2018; Chen et al., 2023; Miao et al., 2023; Xia et al.; Joao Gante, 2023; Yang et al., 2023). However, as far as we know, RaLMSpec is the first work that incorporates the concept of speculative retrieval in RaLM serving and is orthogonal to the speculative inference technique for large language models (LLMs).

## 3 RALMSPEC

A key observation that motivates the design of RaLMSpec is that the *same* document in a knowledge base can be retrieved multiple times during the iterative retrievals of a generative task (e.g., the sequence of the retrieved documents can be $A, B, A, B, C$), also known as the temporal locality in the system domain. Leveraging this observation, RaLMSpec enables a caching-based mechanism for *speculative retrieval*. Combined with *batched verification*, RaLMSpec can thus reduce the serving latency of iterative RaLM while provably preserving its generative quality. We describe the pipeline of RaLMSpec in Algorithm 1.

---

**Algorithm 1** RaLMSpec Pipeline.
___
1: **Input:** Input tokens $X = \{x_0, x_1, \cdots, x_{t-1}\}$, external corpus $\mathcal{C}$, language model $f(\cdot)$
2: **Output:** RaLM generated outputs
3: **Initialize** local cache $Q = \{\}$, speculation stride $s$, model generation stride $k$
4: $q = \text{encode}(X), Q.\text{insert}(\mathcal{C}.\text{retrieve}(q))$          ▷ cache prefetching
5: **while** EOS not in $X$ **do**
6:      **for** $i = 1$ **to** $s$ **do**
7:          $q_i = \text{encode}(X), \hat{d}_i = Q.\text{retrieve}(q_i)$          ▷ speculative retrieval
8:          $\hat{X}_i = f(X, \hat{d}_i, k)$          ▷ model generation step that generates $k$ new tokens
9:          $X = [X, \hat{X}_i]$
10:      **end for**
11:      $d_1, \cdots, d_s = \mathcal{C}.\text{retrieve}(q_1, \cdots, q_s)$          ▷ batched verification
12:      $m = \arg\min_i \hat{d}_i \neq d_i$
13:      **if** $m \leq s$ **then**          ▷ do correction if needed
14:          Roll $X$ back to the $m$-th speculation step
15:          $\hat{X} = f(X, d_i, k)$
16:          $X = [X, \hat{X}]$
17:      **end if**
18: **end while**

---

**Speculative retrieval.** For speculative retrieval, we maintain a local cache for each new request to store retrieved documents. As shown in Figure 2, RaLMSpec utilizes the local cache as a "retrieval" cache instead of a typical exact match cache in the system literature, where a local cache retrieval is performed similarly to a knowledge base retrieval except for the number of documents in the local cache is far less. As there is no entry in the local cache at the start of the process, we perform a retrieval from the knowledge base using the initial query and populate the local cache with the retrieved key and value pairs. The key is usually a vectorized representation of the retrieved documents for dense retrievers or a set of local information (e.g., word level frequency) for sparse

retrievers, while the value is the retrieved documents. For a retrieval step, instead of retrieving from the knowledge base, RaLMSpec retrieves from the local cache speculatively. However, the speculative retrieval results might deviate from the actual retrieval results. To guarantee correctness, a verification step is required for every $s$ consecutive number of speculative retrieval steps performed. We refer to $s$ as the speculation stride. For instance, $s = 3$ in Figure 1(b) for our approach.

A fundamental property that ensures the effectiveness of leveraging a local cache for speculative retrieval is that for most dense and sparse retrievers, relative ranking between documents is preserved between the local cache and the knowledge base. More importantly, if the top-ranked entry in the knowledge base is present in our local cache for a given query, the same entry is guaranteed to be ranked at the top when retrieving from our local cache using the same retrieval metric for this query. For most dense retrievers, this property can be naturally satisfied as the distance metric used by the dense retrievers can be locally computed. For sparse retrievers like BM25 Robertson et al. (2009), we store the corpus-related information throughout the generation process so that the score can be locally computed on the fly. Thus, combined with the temporal locality of the retrieved documents, leveraging a local cache for speculative retrieval can significantly boost the speculation success rate.

**Batched verification.**    During a verification step, we verify the speculated results with a batched retrieval from the knowledge base. For instance, as Figure 1(b) shows, with a speculation stride $s = 3$, the corresponding queries and cache-retrieved documents for the three consecutive speculative retrieval steps (①, ③, ⑤) are $q_0 \rightarrow q_1 \rightarrow q_2$ and A→B→A, respectively. During the verification process, we will retrieve from the external knowledge base with the batched query $\{q_0, q_1, q_2\}$. Suppose the documents retrieved from the knowledge base are $\{A, B, C\}$ for the verification step[1]. We can then validate that the third speculative retrieval step mismatches with the ground truth results. In case of a mismatch, the generation process will roll back to the first mismatch position in the sequence and redo the generation with the correct value (i.e., replacing the third speculated document A with the ground truth document C and continuing generation for the request). In the meantime, we can populate the local cache with the documents retrieved during the verification step. Aside from the *top-1 cache update* approach, RaLMSpec also supports *top-k cache update* as demonstrated in Figure 2. Similar to the concept of prefetching, the top-k cache update aims to fetch more relevant entries to the local cache per verification step to further boost the speculation success rate.

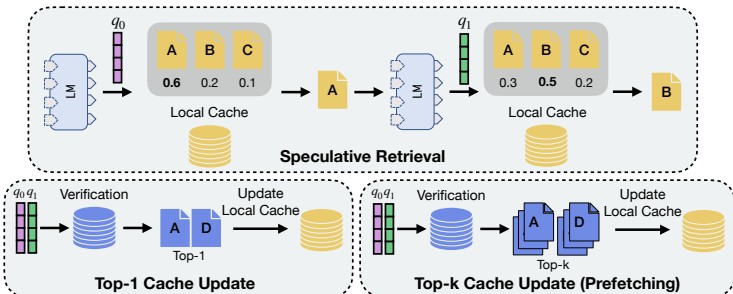

Figure 2: For speculative retrieval, we maintain a local cache for each request and use the same scoring metric as the original retriever to rank the entries within the local cache for a given query. In the verification step, we populate the local cache with either the top-1 or top-k retrieved documents from the knowledge base, where the latter one is referred to as *prefetching*.

Observing that batched retrieval is more efficient by exploiting parallelism, and the speculative retrieval latency is negligible compared to retrieving from the knowledge base. Consequently, in the former toy example, we complete three retrieval steps with only one batched knowledge base retrieval, while a naive implementation requires three knowledge base retrievals. Note that if there is an early mismatch for the speculative retrieval steps, we will incur additional speculation overhead for generating extra tokens. As a result, the speculation stride is a crucial parameter exploiting the trade-off between speculation overhead and retrieval saving. We will elaborate more on choosing an optimal speculation stride $s$ in Section 4.

---

[1]Note that we actually don't need to perform the last speculative retrieval step. We show it in our toy example only to demonstrate the verification process.

In addition to a single-thread implementation, RaLMSpec also considers *asynchronous verification*. Instead of stalling the speculation step during verification, we can launch an extra speculation step asynchronously while the verification of the previous step occurs. This *asynchronous verification* technique is especially beneficial when the verification latency is smaller than the language model's decoding latency as shown in Figure 3. In fact, in this case, *asynchronous verification* with a speculation stride $s = 1$ is the optimal strategy for speculative retrieval. Intuitively, since the verification results can be returned before a speculative retrieval and language model decoding step is completed, we can always hide the latency of a verification step behind a speculation step if the speculation succeeds. More specifically, if the verification succeeds, the model can continue the generation process, saving the verification latency. On the other hand, as soon as the verification fails, the model can regenerate the output based on the corrected information, which falls back to the naive implementation with no speculation overhead.

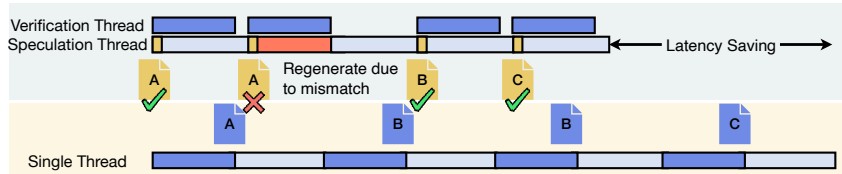

Figure 3: Asynchronous verification obtains latency saving by hiding the verification latency behind a valid speculation step. In case a mismatch is detected between the speculated document and ground truth document, the language model will regenerate outputs using the ground truth document.

## 4 OPTIMAL SPECULATION STRIDE SCHEDULER

RaLMSpec uses *speculation stride s* (defined in Section 3) as a hyperparameter to control the number of speculation steps performed before a verification step. It plays a crucial role in our system due to its effect on the trade-off between speculation overhead and latency saving. Too large a speculation stride would incur high speculation overhead if verification fails early in the stage, while too small a speculation stride does not exploit the benefits of speculation to the fullest. Additionally, depending on different language models, retrieval methods, and speculation accuracy, the optimal choice of the speculation stride varies across diverse setups.

Instead of hand-tuning the speculation stride, we propose the Optimal Speculation Stride Scheduler (OS$^3$), which formalizes this trade-off into an objective function and solves an optimal speculation stride across different configurations adaptively. Given that the goal is to correctly verify a fixed number of documents with minimal latency, we formulate our objective function as the expected number of documents verified successfully per unit time. The goal is therefore to maximize the objective function by optimizing speculation stride $s$.

More precisely, let $a$ denote the latency of a speculation step (speculative retrieval + language model decoding), $b$ denote the latency of a verification step, $d_i$ be the ground truth documents retrieved from the corpus, and $\hat{d}_i$ be the speculated documents retrieved from the local cache. If we define $\gamma(X) = P(d_i = \hat{d}_i \mid X), \forall i \in [s]$ as the speculation accuracy given the current context $X$, then the expected number of verified documents is given by $\frac{1-\gamma(X)^s}{1-\gamma(X)}$ with a speculation stride $s$. We include the derivation in Appendix A.2. For synchronous verification, the latency is calculated as $sa + b$. For asynchronous verification, if all speculated documents match with the ground truth documents in the verification step with a probability of $\gamma(X)^s$, we can benefit from the asynchronous verification with a latency of $(s-1)a + \max(a, b)$. Otherwise, with probability $1 - \gamma(X)^s$ we incur a mismatch so asynchronous verification brings no gain at all with a latency of $sa + b$. Therefore, the expected latency is $[\gamma(X)^s((s-1)a + \max(a, b)) + (1 - \gamma(X)^s)(sa + b)]$ for asynchronous verification. By defining the objective function as $\frac{1-\gamma(X)^s}{(1-\gamma(X))(sa+b)}$ for synchronous verification, or $\frac{1-\gamma(X)^s}{(1-\gamma(X))[\gamma(X)^s((s-1)a+\max(a,b))+(1-\gamma(X)^s)(sa+b)]}$ for asynchronous verification, we can adaptively solve for the optimal $s$ with an estimation of $a, b, \gamma(X)$. Appendix A.2 describes how we estimate $a, b, \gamma(X)$.

## 5 EVALUATION

### 5.1 EXPERIMENTAL SETUPS

We describe our experimental setups in this section, including language models, downstream datasets, retrievers, and the implementation details of the baseline, as well as our approach.

**Language Models.** To demonstrate the effectiveness of our framework with different language models, we select models from three standard natural language generation (NLG) model classes, namely GPT2, OPT, and LLaMA-2 (Radford et al., 2019; Zhang et al., 2022; Touvron et al., 2023). More specifically, we choose GPT2-medium, OPT-1.3B, and LLaMA-2-7B, which are commonly used as base language models in RaLM and, at the same time, span across different model sizes.

**Datasets.** For the downstream workload, we mainly focus on the knowledge-intensive open-domain question-answering tasks. We thus include four QA datasets in our experiments: Wiki-QA, Web Questions, Natural Question, and Trivia-QA (Yang et al., 2015; Berant et al., 2013; Kwiatkowski et al., 2019; Joshi et al., 2017). For all tasks, we use the Wikipedia corpus as our external knowledge base (Chen et al., 2017).

**Retrievers.** To demonstrate the consistency of our approach, we test our method against both dense retrievers (vector-based) and sparse retrievers (bag-of-words-based). For dense retrievers, we further experiment with the exact and approximate methods, where the approximate method is much faster but less accurate. We use the Dense Passage Retriever (DPR) (Karpukhin et al., 2020) as the exact dense retriever (EDR), and its approximate version DPR-HNSW as the approximate dense retriever (ADR) (Malkov & Yashunin, 2018). For the sparse retriever (SR), we use the BM25 retriever (Robertson et al., 2009). We use the implementation from Pyserini (Lin et al., 2021) for all dense and sparse retrievers, where the dense retrievers are built on top of the standard FAISS library (Johnson et al., 2019).

**Baseline.** For baseline implementation, we follow directly from the iterative RaLM implementation as in Ram et al. (2023), where retrieval is triggered every four tokens generated by the language model. The latest retrieved document chunk is directly prepended to the prompt and replaces previous documents. We use RaLMSeq to denote the baseline implementation in the rest of the paper.

**Implementation Details.** For both RaLMSpec and RaLMSeq, we set the maximum input prompt length to be 512 tokens and the maximum generation length to be 128 tokens. The maximum length of the retrieved document chunk is set to 256 as in Ram et al. (2023). For naive RaLMSpec, we use a constant speculation stride $s = 3$. Whenever $OS^3$ is enabled, we initialize the speculation stride with $s = 1$ and let the scheduler adapt onwards. In all our experiments, we set the window size $w = 5$ and $\gamma_{max} = 0.6$ for estimating $\gamma$. For prefetching, we use a prefetch size of 20. We also test with a prefetch size of 256 for the ablation study. Due to the existence of Global Interpreter Lock (GIL) in Python, the potential of asynchronous verification cannot be fully realized. Thus, for asynchronous verification only, we use a simulated latency by calculating the ideal running time without additional overhead[2]. Except for asynchronous verification, all latencies are measured in wall-clock time. We want to note that asynchronous verification is not a significant factor contributing to the speed-up; details are provided in the ablation study in Section 5.2. Both RaLMSeq and RaLMSpec are written in Python and tested on the VM.GPU.A10 instance on the Oracle cloud, which contains one A10 GPU and 15 CPUs.

### 5.2 RESULTS

In this section, we empirically verify our approach against diverse language models, retrievers, and downstream datasets. We demonstrate our main results in Figure 4. RaLMSpec+P denotes RaLMSpec with prefetching enabled, RaLMSpec+S denotes RaLMSpec with the optimal speculation stride scheduler ($OS^3$) enabled, and RaLMSpec+A denotes RaLMSpec with asynchronous verification enabled. RaLMSpec+PSA indicates RaLMSpec with all three components enabled. For each dataset, we randomly select 100 questions to test the latency results with RaLMSeq and RaLMSpec. To show the confidence interval, we further plot the mean and standard deviation over five independent runs for each setup. With the exact dense retriever (EDR), RaLMSpec+PSA can reduce the latency by $2.39\times$ for GPT2, $2.33\times$ for OPT, and $1.75\times$ for LLaMA-2 compared with

---

[2]Enabling an asynchronous verification only requires two parallel threads (one thread for model generation and one thread for retrieval), thus the overhead should be minimal.

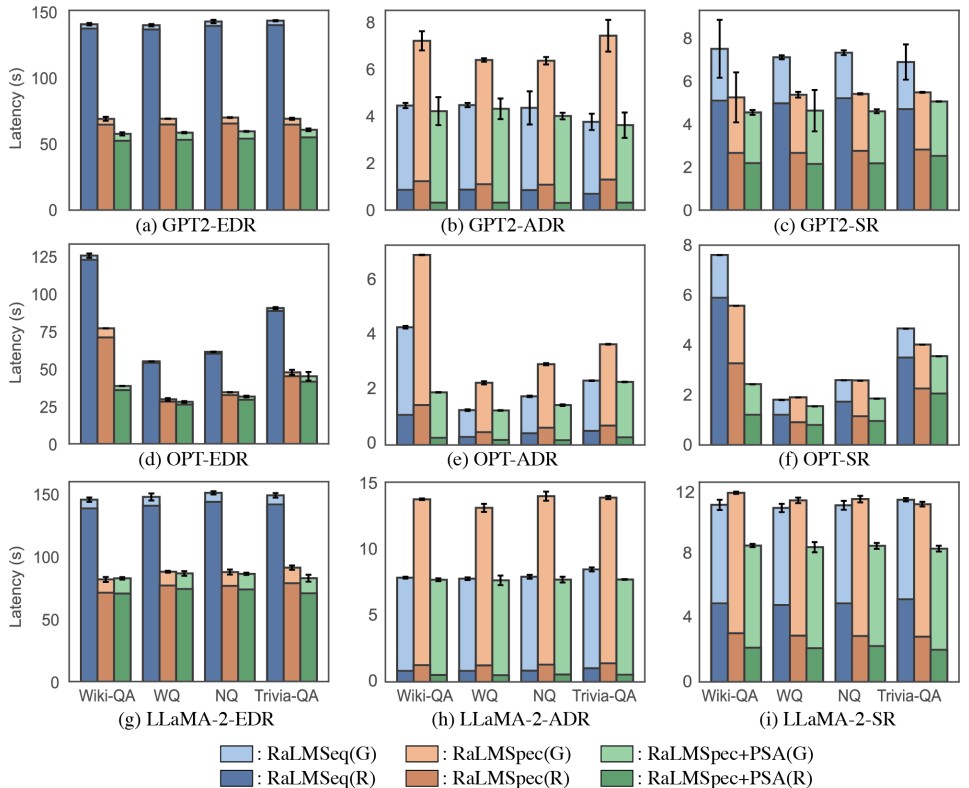

Figure 4: Latency comparison between RaLMSeq, RaLMSpec, and RaLMSpec+PSA on GPT2-medium, OPT-1.3B, and LLaMA-2-7B over four QA datasets with three different types of retrievers, where EDR, ADR, SR stand for exact dense retriever, approximate dense retriever, and sparse retriever respectively. We decompose the overall latency into the language model generation latency (G) and retrieval latency (R) to demonstrate the trade-off.

RaLMSeq. With the approximate dense retriever (ADR), RaLMSpec+PSA can reduce the latency by $1.05\times$ for GPT2, $1.39\times$ for OPT, and $1.04\times$ for LLaMA-2 compared with RaLMSeq. With the sparse retriever (SR), RaLMSpec+PSA can reduce the latency by $1.53\times$ for GPT2, $1.77\times$ for OPT, and $1.31\times$ for LLaMA-2 compared with RaLMSeq. We include the full results in Appendix A.7.

**Analysis.** As shown in Figure 4, RaLMSpec+PSA achieves the best performance consistently across all scenarios. However, the most significant speed-up ratio is achieved when the retriever is the exact dense retriever. This is because our approach can only optimize for the retrieval latency (R), not the language model generation latency (G). Thus, our speed-up is intrinsically bottlenecked by the ratio of the retrieval latency over the end-to-end latency. When we use the approximate dense or sparse retriever, the naive implementation of RaLMSpec performs worse than the baseline for some cases. This relates to a constant speculation stride being used, and consequently, the trade-off between the speculation overhead and gain might not be optimal. More specifically, as the language model decoding step latency outweighs the retrieval latency, being too optimistic in the speculation stride can result in excessive speculation overhead due to an early mismatch in the verification step. On the other hand, if we enable the optimal speculation stride scheduler ($OS^3$), we can resolve this issue as the speculation stride can then be optimally adapted. Combined with prefetching and asynchronous verification, RaLMSpec+PSA achieves the best speed-up ratio performance consistently and automatically across all scenarios.

**Ablation study.** We further present the contribution of each component (prefetching, $OS^3$, and asynchronous verification) in Table 1 on GPT2, OPT, and LLaMA-2 with the exact dense retriever (EDR), approximate dense retriever (ADR), and sparse retriever (SR). The speed-up ratio is compared against the baseline (RaLMSeq) and averaged over the four datasets. In most cases, enabling $OS^3$ brings the most significant gain among the three components, while combining all three elements achieves the best performance consistently. This reflects that controlling the trade-off between

Table 1: Ablation results of speed-up ratio compared with baseline of each component. P stands for prefetching, S stands for optimal speculation stride scheduler and A stands for asynchronous verification. $(*)$ and $(**)$ denote the most speed-up and the second most speed-up respectively.

| Retriever | Method | GPT2 | OPT | LLaMA-2 |
|---|---|---|---|---|
| EDR | RaLMSpec | $2.04\times$ | $1.76\times$ | $1.70\times$ |
| | RaLMSpec+P | $2.10\times$ | $2.16\times(**)$ | $1.75\times(**)$ |
| | RaLMSpec+S | $2.26\times(**)$ | $2.15\times$ | $1.69\times$ |
| | RaLMSpec+A | $2.03\times$ | $1.74\times$ | $1.74\times$ |
| | RaLMSpec+PSA | $2.39\times(*)$ | $2.32\times(*)$ | $1.75\times(*)$ |
| ADR | RaLMSpec | $0.62\times$ | $0.61\times$ | $0.58\times$ |
| | RaLMSpec+P | $0.59\times$ | $0.76\times$ | $0.58\times$ |
| | RaLMSpec+S | $0.92\times(**)$ | $1.17\times(**)$ | $1.01\times(**)$ |
| | RaLMSpec+A | $0.66\times$ | $0.46\times$ | $0.55\times$ |
| | RaLMSpec+PSA | $1.05\times(*)$ | $1.39\times(*)$ | $1.04\times(*)$ |
| SR | RaLMSpec | $1.34\times$ | $1.18\times$ | $0.97\times$ |
| | RaLMSpec+P | $1.39\times$ | $1.42\times$ | $0.98\times$ |
| | RaLMSpec+S | $1.32\times$ | $1.52\times(**)$ | $1.05\times(**)$ |
| | RaLMSpec+A | $1.41\times(**)$ | $1.27\times$ | $1.01\times$ |
| | RaLMSpec+PSA | $1.53\times(*)$ | $1.77\times(*)$ | $1.31\times(*)$ |

speculation overhead and latency reduction with the speculation stride is critical for achieving the optimal speed-up under various scenarios. The results demonstrate that $OS^3$ can find a better stride scheduling solution than the naive hand-tuned constant speculation stride in most cases. Prefetching can also improve performance by caching more entries in the local cache to obtain a higher speculation accuracy. However, increasing a prefetching size can introduce higher retrieval overhead. As shown in Table 2, when we increase the prefetching size from 20 to 256, the performance decreases in most cases due to the diminished prefetching gain and increased retrieval overhead. Asynchronous verification improves the performance by introducing an extra speculation step when doing verification. However, if the verification fails at an earlier stage, the benefit of doing asynchronous verification cannot be realized. As prefetching, $OS^3$, and asynchronous verification compensate one another, combining all of them fully realizes our approach's potential.

Table 2: Ablation results of speed-up ratio of different prefetching size compared with the baseline.

| Retriever | Method | GPT2 | OPT | LLaMA-2 |
|---|---|---|---|---|
| EDR | RaLMSpec+P(20) | $2.10\times$ | $\mathbf{2.16\times}$ | $\mathbf{1.75\times}$ |
| | RaLMSpec+P(256) | $\mathbf{2.15\times}$ | $1.72\times$ | $1.63\times$ |
| ADR | RaLMSpec+P(20) | $0.59\times$ | $\mathbf{0.76\times}$ | $\mathbf{0.58\times}$ |
| | RaLMSpec+P(256) | $\mathbf{0.67\times}$ | $0.25\times$ | $0.34\times$ |
| SR | RaLMSpec+P(20) | $\mathbf{1.39\times}$ | $\mathbf{1.42\times}$ | $\mathbf{0.98\times}$ |
| | RaLMSpec+P(256) | $1.02\times$ | $0.93\times$ | $0.84\times$ |

## 6 CONCLUSION

In this work, we introduce RaLMSpec, a speculation-inspired framework that accelerates the serving of generic retrieval augmented generation approaches that suffer from frequent retrieval-generation interactions. By leveraging the temporal locality of retrieved documents, we enable a request-level local cache for speculative retrieval and a batched verification step to guarantee correctness. In addition, we introduce cache prefetching, an optimal speculation stride scheduler, and asynchronous verification to boost the speculation performance further. The effectiveness of RaLMSpec has been verified empirically against different language models, retrievers, and downstream datasets. The results demonstrate that RaLMSpec can indeed have substantial speed-ups consistently in all scenarios compared with the baseline.

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

## A APPENDIX

### A.1 BATCHED RETRIEVAL

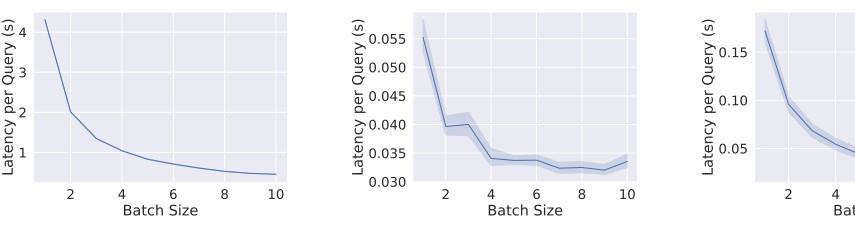

(a) Exact dense retriever.    (b) Approximate dense retriever    (c) Sparse retriever

Figure 5: Effect of batch size on latency per query for three different types of retrievers. 95% confidence bands for the true mean latency are included.

We demonstrate the benefit of batched retrievals by examining the latency per query for increasing batch sizes. For all three retrieval methods, latency per query decreases with increasing batch size. This is most noticeable with the exact dense retriever and sparse retriever, where total retrieval time was essentially constant across all batch sizes. The approximate dense retriever exhibited latency that scaled linearly with batch size; however, there was a significant intercept term in the linear relationship. With batch retrievals, repeated incurrences of this latency under individual retrievals are saved. The result is less latency per query for approximate dense retrievers as well, though not to the extent of exact dense retrievers or sparse retrievers.

### A.2 DETAILED DERIVATIONS

**Expected matching length.** Given a speculation accuracy of $\gamma(X) \in [0, 1]$ for single-step speculation, the expected number of matched documents with a speculation stride $s$ can be calculated as:

$$
\begin{aligned}
\mathbb{E}\left[\# \text{ of verified documents} \mid X, s\right] &= 1 + \sum_{i=1}^{s-2} i\gamma(X)^i(1 - \gamma(X)) + (s-1)\gamma(X)^{(s-1)} \\
&= 1 + \sum_{i=1}^{s-2} i\gamma(X)^i - \sum_{i=1}^{s-1}(i-1)\gamma(X)^i + (s-1)\gamma(X)^{(s-1)} \\
&= \sum_{i=0}^{s-1} \gamma(X)^i \\
&= \frac{1 - \gamma(X)^s}{1 - \gamma(X)}
\end{aligned}
$$

**Parameter estimation for OS³** To get an optimal stride $s$, we need to adaptively estimate $a, b, \gamma(X)$. For $a, b$, we directly estimate their value with the profiling results from the most recent steps. For the exact dense retriever and sparse retriever, we observe that the latency of batched retrieval with a batch size smaller than 10 is nearly constant, while for the approximate dense retriever, the latency is surprisingly linear to the batch size but with a large intercept (batch retrieval is still more efficient due to the intercept is non-zero). We include the batched retrieval latency analysis for three different retrievers in Appendix A.1. For $\gamma(X)$, we use the maximum log-likelihood estimation within a specific window size $w$ so that the estimated $\hat{\gamma}$ can have both locality and less variance. More specifically, denoting $s(t), t \in [w]$ as the speculation stride (also the batch size) in the $t$-th most recent verification step, $M(s(t), X)$ as the corresponding number of matched documents, we estimate $\gamma(X)$ with

$$
\hat{\gamma}(X) = \frac{\sum_t M(s(t), X)}{\sum_t M(s(t), X) + \sum_t \mathbb{1}(M(s(t), X) < s(t))}
$$

where $\mathbb{1}(\cdot)$ is the indicator function. To prevent the over-optimistic estimation and division-by-zero error when $\hat{\gamma}$ approaches probability 1 in some cases, we further set a constant upper bound $\gamma_{\max}$ and truncate $\hat{\gamma}$ accordingly.

A.3 KNN-LM Serving Evaluation

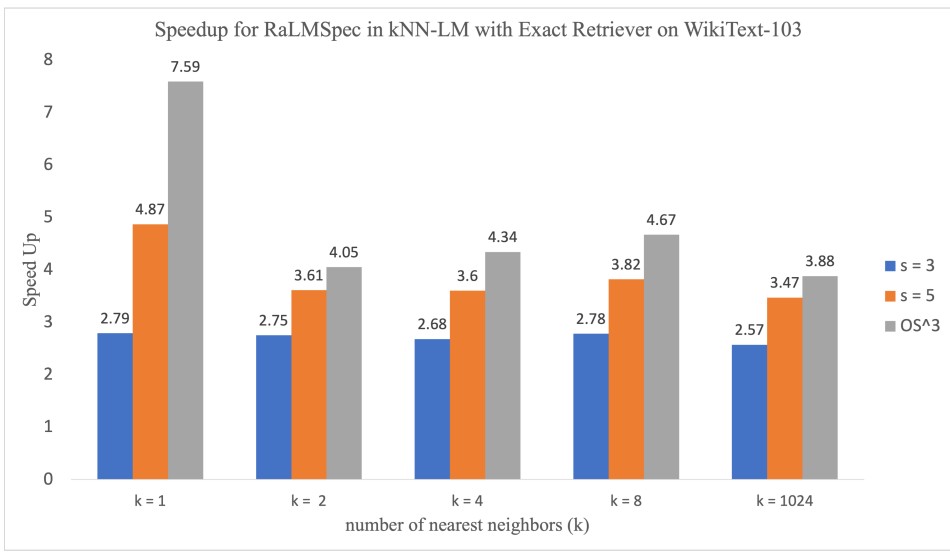

(a) Exact dense retriever.

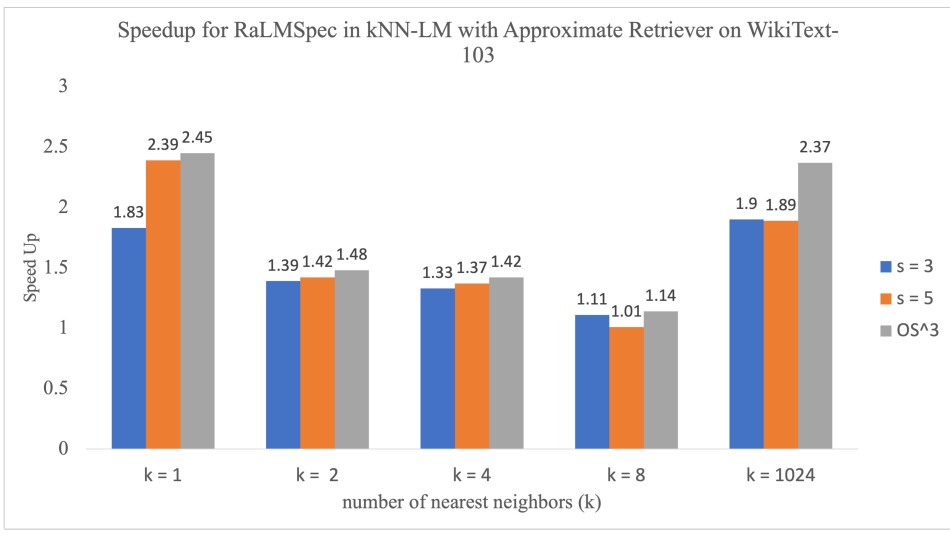

(b) Approximate dense retriever

Figure 6: Speedup Ratio Results for RaLMSpec on kNN-LMs using Wikipedia-QA. k stands for number of nearest neighbors in kNN-LMs, s stands for stride size, OS3 stands for optimal scheduler stride.

We further evaluate our approach against a retrieval-intensive workload KNN-LM (Khandelwal et al., 2019). For KNN-LM, the knowledge base is constructed for each training token, with the key being the embedding of its leftward context and the value being the token itself. Instead of relying on a single language model to generate the next token sampling distribution, KNN-LM retrieves $k$ closest entries in a knowledge base along with their target tokens using an embedding of the current context. A distribution over these $k$ target tokens is then computed based on their distance with respect to the current context embedding. The distribution is then interpolated with the original language model distribution to get the next token sampling distribution. Khandelwal et al. (2019) indicates KNN-LM can improve the perplexity of the base language model to state-of-the-art without extra training. While effective, the inference overhead of KNN-LM models is prohibitive as retrieval will be performed for every token generation step. To this end, we are interested in testing our system against the KNN-LM models. Different from the original speculation cache design, we cannot populate the cache by adding the same documents, as it will likely not be retrieved again in

future decoding steps. However, for speculation, we can populate the cache with the next $n$ entries directly after the currently retrieved item. In addition, instead of enforcing all retrieved items to be the same for the verification step, we identify that equivalency can be preserved as long as the speculated next token matches with the ground truth next token. This relaxation is critical when $k$ is large, e.g., $k = 1024$. Matching all 1024 entries with the ground truth one is exponentially hard, but matching the ground truth decoded token can be easier. By modifying the cache update rule and verification protocol as above, we can achieve significant speed-up ratios compared with the naive implementation as shown in Figure 6. We have verified our approach against different $k$ values ranging from 1 to 1024. When the retriever is an exact dense retriever, RaLMSpec can achieve up to $7.59\times$ acceleration rate with the optimal speculation stride scheduler ($OS^3$) and even $3.88\times$ acceleration rate when $k = 1024$. When the retriever is an approximate dense retriever, RaLMSpec can achieve up to $2.45\times$ acceleration rate with the optimal speculation stride scheduler ($OS^3$) and even $2.37\times$ acceleration rate when $k = 1024$. We have further experimented with different speculation stride sizes to have some ablation studies on the importance of the stride. The results show that a larger stride is better for the exact dense retriever, while a smaller stride is more suitable for the approximate dense retriever when $k = 8, 1024$. Enabling the optimal speculation stride scheduler can achieve the best performance consistently across all scenarios.

## A.4 LLaMA-2-13B Serving Evaluation

Table 3: RaLMSpec+PSA speed-up ratio compared against the baseline on LLaMA-2-13B over four downstream datasets (Wiki QA, Web Questions, Natural Questions, and Trivia-QA).

| Retriever | Wiki QA | Web Questions | Natural Questions | Trivia-QA |
|-----------|---------|---------------|-------------------|-----------|
| EDR | $1.70\times$ | $1.85\times$ | $1.73\times$ | $1.78\times$ |
| ADR | $1.03\times$ | $1.04\times$ | $1.02\times$ | $1.03\times$ |
| SR | $1.18\times$ | $1.21\times$ | $1.22\times$ | $1.26\times$ |

To demonstrate the effectiveness of RaLMSpec, we evaluate RaLMSpec+PSA with the LLaMA-2-13B model over the Wiki QA, Web Questions, Natural Questions, and Trivia-QA and present the results in Table 3. We can still observe RaLMSpec+PSA can achieve up to $1.85\times$ speed-up ratio compared against RaLMSeq when the retriever is the exact dense retriever. We observe marginal improvement for the approximate dense retriever because language model generation latency has far outweighed the retrieval latency. Thus, the retrieval latency saving is amortized within the end-to-end latency saving.

## A.5 Ablation Study on Different System Components

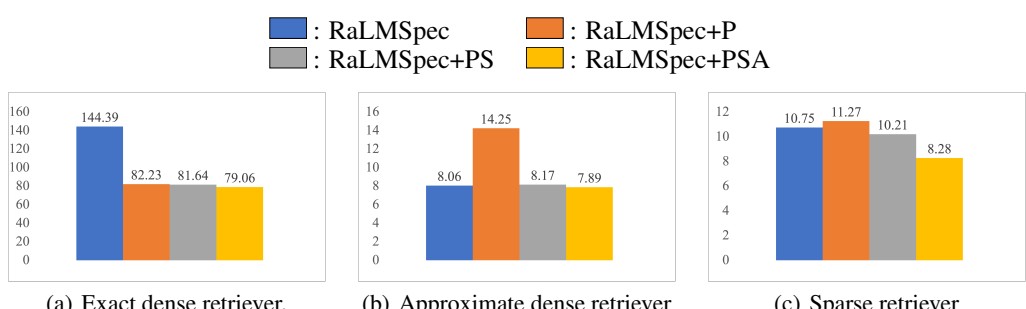

(a) Exact dense retriever.    (b) Approximate dense retriever    (c) Sparse retriever

Figure 7: Ablation study on the contribution of the prefetching (P), optimal speculation stride scheduler (S), and asynchronous verification (A) components.

Table 4: Speed-up contribution of different combinations of prefetching (P), optimal speculation stride scheduler (S), and asynchronous verification (A). We report the average serving latency over 100 requests evaluated over LLaMA-2-7B and the Wiki QA dataset.

| Retriever | B | P | S | A | PS | SA | PA | PSA |
|-----------|---|---|---|---|----|----|----|----|
| EDR | 144.39s | 82.23s | 85.19s | 90.49s | 81.64s | 85.13s | **81.60s** | **79.06s** |
| ADR | 8.06s | 14.25s | 8.14s | 13.90s | 8.17s | **7.83s** | 12.84s | **7.89s** |
| SR | 10.75s | 11.27s | 10.38s | 10.88s | 10.21s | **8.26s** | 10.61s | **8.28s** |

To demonstrate the contribution of different components, we have evaluated all combinations among prefetching (P), optimal speculation stride scheduler (S), and asynchronous verification (A) with the LLaMA-2-7B model on the Wiki QA dataset and presented the results in Table 4 and Figure 7. When the retriever is the exact dense retriever, prefetching can improve more effectively than the optimal speculation stride scheduler and asynchronous verification. In addition, prefetching+asynchronous verification can even outperform RaLMSpec+PSA. This is due to the prefixed stride (s=3) for the exact dense retriever is already near-optimal, and the optimal speculation stride scheduler needs a warm-up phase at the start to adapt to the optimal stride, thus incurring some additional overheads. We can observe that the optimal speculation stride scheduler is the most crucial component for the approximate dense and sparse retriever and achieves the best performance when combined with asynchronous verification. Prefetching can even affect performance negatively because the overhead of prefetching outweighs its gain. To sum up, enabling the optimal speculation stride scheduler for all workloads can help achieve a near-optimal stride adaptively. For workloads where retrieval dominates most of the latency, enabling prefetching can further improve the serving latency while the contribution of asynchronous verification is marginal. For workloads where retrieval is not the most time-consuming step, asynchronous verification can benefit the serving latency while prefetching's overhead can outweigh its benefit if the prefetching size has not been chosen properly.

## A.6 ABLATION STUDY ON SPECULATION STRIDE

Table 5: Ablation results on different speculation strides (S=2,4,8) and the optimal speculation stride scheduler (OS$^3$). We report the average serving latency over 100 requests evaluated over LLaMA-2-7B and the Wiki QA dataset.

| Retriever | S=2 | S=4 | S=8 | OS$^3$ |
|-----------|-----|-----|-----|--------|
| EDR | 92.17s | **81.06s** | 81.90s | 85.19s |
| ADR | 9.86s | 14.93s | 25.88s | **8.14s** |
| SR | 10.65s | 12.48s | 16.66s | **10.38s** |

To demonstrate the effect of the optimal speculation stride scheduler, we have conducted ablation studies on different fixed speculation strides (s=2, 4, 8) with LLaMA-2-7B over the Wiki QA dataset. The results are presented in Table 5. We can observe from the results that a larger speculation stride is better when the exact dense retriever is used. This is because the retrieval latency is much larger for the exact dense retriever and way exceeds the language model generation latency. Thus, an aggressive speculation stride will incur little overhead while having the opportunity for a longer match. On the other hand, the approximately dense and sparse retrievers prefer a smaller speculation stride because the retrieval is more efficient, and the cost-performance ratio for doing more speculations is low. Thus, the optimal speculation stride scheduler is designed to tackle this dynamism and can achieve near-optimal stride scheduling in all scenarios. OS$^3$ performs slightly worse than $s = 4, 8$ in the case of EDR because a warm-up phase is required for OS$^3$ to start its adaptation, i.e., we initialize s=1 when we enable OS$^3$. Thus, the warm-up phase can introduce non-optimal strides and, thus, slightly worse performance.

## A.7 ADDITIONAL EXPERIMENTAL RESULTS

This section presents the full evaluation results with GPT2, OPT, and LLaMA-2 over Wiki QA, Web Questions, Natural Questions, and Trivia-QA against the exact dense, approximate dense, and sparse retriever.

Table 6: Averaged latency measured in seconds over GPT-2. OPT and LLaMA-2 with the exact dense retriever.

| Model | Method | Wiki QA | WQ | NQ | Trivia QA |
|---|---|---|---|---|---|
| GPT2 | Baseline | $142.14 \pm 0.96$ | $141.38 \pm 1.50$ | $144.22 \pm 1.17$ | $144.82 \pm 0.96$ |
| | RaLMSpec | $69.82 \pm 0.22$ | $69.88 \pm 0.52$ | $70.79 \pm 1.27$ | $69.83 \pm 0.28$ |
| | RaLMSpec+P(20) | $68.22 \pm 0.18$ | $68.09 \pm 0.18$ | $68.06 \pm 0.37$ | $68.14 \pm 0.06$ |
| | RaLMSpec+P(256) | $66.14 \pm 0.39$ | $65.22 \pm 0.79$ | $67.10 \pm 0.59$ | $66.64 \pm 0.23$ |
| | RaLMSpec+S | $62.72 \pm 0.48$ | $62.43 \pm 0.19$ | $63.48 \pm 0.69$ | $64.63 \pm 0.44$ |
| | RaLMSpec+A | $69.92 \pm 1.06$ | $69.36 \pm 0.60$ | $71.00 \pm 0.74$ | $70.40 \pm 0.78$ |
| | RaLMSpec+P(20)SA | $58.35 \pm 0.31$ | $59.24 \pm 0.46$ | $60.21 \pm 0.78$ | $61.39 \pm 0.99$ |
| | RaLMSpec+P(256)SA | $\mathbf{53.95 \pm 0.72}$ | $\mathbf{53.36 \pm 1.08}$ | $\mathbf{56.26 \pm 0.95}$ | $\mathbf{56.95 \pm 1.34}$ |
| OPT | Baseline | $126.86 \pm 1.39$ | $55.60 \pm 0.08$ | $62.02 \pm 0.11$ | $91.50 \pm 0.01$ |
| | RaLMSpec | $77.81 \pm 0.84$ | $29.99 \pm 0.54$ | $34.75 \pm 0.19$ | $48.20 \pm 0.09$ |
| | RaLMSpec+P(20) | $40.37 \pm 0.07$ | $29.09 \pm 0.07$ | $33.79 \pm 0.60$ | $52.09 \pm 0.11$ |
| | RaLMSpec+P(256) | $72.68 \pm 0.75$ | $31.94 \pm 1.16$ | $39.90 \pm 3.18$ | $50.24 \pm 0.02$ |
| | RaLMSpec+S | $40.77 \pm 0.52$ | $29.49 \pm 0.53$ | $35.13 \pm 0.10$ | $50.82 \pm 0.49$ |
| | RaLMSpec+A | $77.76 \pm 4.99$ | $30.28 \pm 0.59$ | $36.16 \pm 1.18$ | $47.83 \pm 0.03$ |
| | RaLMSpec+P(20)SA | $\mathbf{39.00 \pm 0.58}$ | $28.31 \pm 0.62$ | $31.88 \pm 1.67$ | $45.51 \pm 2.93$ |
| | RaLMSpec+P(256)SA | $59.21 \pm 0.04$ | $\mathbf{27.79 \pm 0.11}$ | $\mathbf{30.02 \pm 0.01}$ | $\mathbf{45.13 \pm 0.04}$ |
| LLaMA | Baseline | $144.39 \pm 1.71$ | $146.52 \pm 1.92$ | $149.76 \pm 0.95$ | $147.76 \pm 2.80$ |
| | RaLMSpec | $81.05 \pm 0.78$ | $87.20 \pm 1.83$ | $86.92 \pm 1.30$ | $90.44 \pm 2.01$ |
| | RaLMSpec+P(20) | $83.94 \pm 1.11$ | $\mathbf{84.23 \pm 0.37}$ | $84.74 \pm 0.47$ | $81.86 \pm 0.76$ |
| | RaLMSpec+P(256) | $82.23 \pm 1.95$ | $94.15 \pm 1.60$ | $97.14 \pm 1.89$ | $85.65 \pm 1.46$ |
| | RaLMSpec+S | $85.19 \pm 2.26$ | $88.95 \pm 0.99$ | $89.45 \pm 1.28$ | $84.28 \pm 2.93$ |
| | RaLMSpec+A | $90.49 \pm 6.12$ | $85.74 \pm 1.94$ | $\mathbf{84.37 \pm 0.62}$ | $77.39 \pm 1.11$ |
| | RaLMSpec+P(20)SA | $81.94 \pm 0.91$ | $85.81 \pm 1.82$ | $85.47 \pm 1.70$ | $82.03 \pm 2.73$ |
| | RaLMSpec+P(256)SA | $\mathbf{79.06 \pm 3.61}$ | $87.34 \pm 4.63$ | $95.54 \pm 3.36$ | $\mathbf{73.64 \pm 0.80}$ |

Table 7: Averaged latency measured in seconds over GPT-2, OPT and LLaMA-2 with the approximate dense retriever.

| Model | Method | Wiki QA | WQ | NQ | Trivia QA |
|---|---|---|---|---|---|
| GPT2 | Baseline | $4.48 \pm 0.11$ | $4.50 \pm 0.41$ | $4.38 \pm 0.60$ | $3.78 \pm 0.10$ |
| | RaLMSpec | $7.26 \pm 0.07$ | $6.44 \pm 0.44$ | $6.41 \pm 0.71$ | $7.47 \pm 0.16$ |
| | RaLMSpec+P(20) | $6.92 \pm 0.10$ | $7.38 \pm 0.56$ | $7.37 \pm 0.58$ | $7.28 \pm 0.28$ |
| | RaLMSpec+P(256) | $6.65 \pm 0.07$ | $5.97 \pm 0.46$ | $5.64 \pm 0.74$ | $6.96 \pm 0.63$ |
| | RaLMSpec+S | $4.59 \pm 0.28$ | $4.77 \pm 0.32$ | $4.65 \pm 0.61$ | $4.51 \pm 0.45$ |
| | RaLMSpec+A | $6.50 \pm 0.54$ | $6.49 \pm 0.38$ | $5.70 \pm 0.70$ | $6.94 \pm 0.84$ |
| | RaLMSpec+P(20)SA | $4.24 \pm 0.14$ | $4.34 \pm 0.35$ | $4.03 \pm 0.68$ | $\mathbf{3.64 \pm 0.54}$ |
| | RaLMSpec+P(256)SA | $\mathbf{4.01 \pm 0.21}$ | $\mathbf{3.81 \pm 0.02}$ | $\mathbf{3.40 \pm 0.01}$ | $3.86 \pm 0.31$ |
| OPT | Baseline | $4.43 \pm 0.05$ | $1.31 \pm 0.01$ | $1.83 \pm 0.01$ | $2.42 \pm 0.03$ |
| | RaLMSpec | $7.15 \pm 0.06$ | $2.34 \pm 0.01$ | $3.04 \pm 0.03$ | $3.79 \pm 0.04$ |
| | RaLMSpec+P(20) | $3.44 \pm 0.02$ | $2.34 \pm 0.01$ | $2.70 \pm 0.06$ | $4.66 \pm 0.03$ |
| | RaLMSpec+P(256) | $16.03 \pm 0.03$ | $6.06 \pm 0.01$ | $7.06 \pm 0.04$ | $10.17 \pm 0.01$ |
| | RaLMSpec+S | $2.21 \pm 0.01$ | $1.47 \pm 0.01$ | $1.88 \pm 0.05$ | $2.97 \pm 0.05$ |
| | RaLMSpec+A | $7.55 \pm 0.05$ | $2.25 \pm 0.01$ | $5.41 \pm 1.32$ | $6.20 \pm 0.02$ |
| | RaLMSpec+P(20)SA | $\mathbf{1.98 \pm 0.03}$ | $\mathbf{1.30 \pm 0.01}$ | $\mathbf{1.50 \pm 0.01}$ | $\mathbf{2.37 \pm 0.01}$ |
| | RaLMSpec+P(256)SA | $9.41 \pm 0.66$ | $4.14 \pm 0.02$ | $4.31 \pm 0.02$ | $6.19 \pm 0.02$ |
| LLaMA | Baseline | $8.06 \pm 0.07$ | $7.97 \pm 0.06$ | $8.11 \pm 0.11$ | $8.68 \pm 0.10$ |
| | RaLMSpec | $14.10 \pm 0.31$ | $13.44 \pm 0.37$ | $14.35 \pm 0.15$ | $14.23 \pm 0.35$ |
| | RaLMSpec+P(20) | $14.25 \pm 0.39$ | $13.45 \pm 0.28$ | $14.08 \pm 0.32$ | $14.21 \pm 0.30$ |
| | RaLMSpec+P(256) | $20.63 \pm 0.48$ | $26.44 \pm 3.11$ | $27.38 \pm 3.39$ | $21.04 \pm 0.43$ |
| | RaLMSpec+S | $8.14 \pm 0.19$ | $8.08 \pm 0.07$ | $8.08 \pm 0.07$ | $8.16 \pm 0.09$ |
| | RaLMSpec+A | $13.90 \pm 0.36$ | $13.28 \pm 0.17$ | $13.72 \pm 0.14$ | $18.35 \pm 1.11$ |
| | RaLMSpec+P(20)SA | $\mathbf{7.89 \pm 0.22}$ | $\mathbf{7.84 \pm 0.15}$ | $\mathbf{7.90 \pm 0.12}$ | $\mathbf{7.91 \pm 0.03}$ |
| | RaLMSpec+P(256)SA | $14.06 \pm 0.08$ | $14.96 \pm 1.34$ | $14.59 \pm 2.04$ | $12.94 \pm 0.03$ |

Table 8: Averaged latency measured in seconds over GPT-2. OPT and LLaMA-2 with the sparse retriever.

| Model | Method | Wiki QA | WQ | NQ | Trivia QA |
|---|---|---|---|---|---|
| GPT2 | Baseline | $7.41 \pm 1.34$ | $7.03 \pm 1.15$ | $7.23 \pm 0.11$ | $6.80 \pm 0.09$ |
| | RaLMSpec | $5.18 \pm 0.13$ | $5.30 \pm 0.95$ | $5.34 \pm 0.11$ | $5.40 \pm 0.03$ |
| | RaLMSpec+P(20) | $5.23 \pm 0.23$ | $4.58 \pm 0.01$ | $5.17 \pm 0.05$ | $5.50 \pm 0.04$ |
| | RaLMSpec+P(256) | $6.88 \pm 0.66$ | $7.16 \pm 1.34$ | $6.76 \pm 0.27$ | $6.91 \pm 0.13$ |
| | RaLMSpec+S | $5.62 \pm 0.96$ | $5.03 \pm 0.68$ | $5.24 \pm 0.13$ | $5.61 \pm 0.11$ |
| | RaLMSpec+A | $5.34 \pm 0.89$ | $4.99 \pm 0.86$ | $4.76 \pm 0.14$ | $5.04 \pm 0.12$ |
| | RaLMSpec+P(20)SA | $\mathbf{4.49 \pm 0.09}$ | $\mathbf{4.57 \pm 0.81}$ | $\mathbf{4.54 \pm 0.02}$ | $\mathbf{4.99 \pm 0.01}$ |
| | RaLMSpec+P(256)SA | $6.66 \pm 1.25$ | $6.50 \pm 1.39$ | $5.54 \pm 0.02$ | $5.91 \pm 0.03$ |
| OPT | Baseline | $7.68 \pm 0.01$ | $1.83 \pm 0.01$ | $2.62 \pm 0.01$ | $4.71 \pm 0.02$ |
| | RaLMSpec | $5.63 \pm 0.01$ | $1.93 \pm 0.01$ | $2.60 \pm 0.01$ | $4.07 \pm 0.02$ |
| | RaLMSpec+P(20) | $3.00 \pm 0.01$ | $2.06 \pm 0.01$ | $2.50 \pm 0.02$ | $4.27 \pm 0.01$ |
| | RaLMSpec+P(256) | $7.13 \pm 0.01$ | $2.45 \pm 0.01$ | $3.22 \pm 0.01$ | $5.24 \pm 0.02$ |
| | RaLMSpec+S | $2.79 \pm 0.01$ | $1.69 \pm 0.01$ | $2.32 \pm 0.01$ | $4.27 \pm 0.01$ |
| | RaLMSpec+A | $5.27 \pm 0.02$ | $1.86 \pm 0.01$ | $2.28 \pm 0.01$ | $3.82 \pm 0.02$ |
| | RaLMSpec+P(20)SA | $\mathbf{2.46 \pm 0.01}$ | $\mathbf{1.57 \pm 0.01}$ | $\mathbf{1.88 \pm 0.01}$ | $\mathbf{3.59 \pm 0.01}$ |
| | RaLMSpec+P(256)SA | $6.37 \pm 0.01$ | $1.92 \pm 0.04$ | $2.44 \pm 0.02$ | $4.53 \pm 0.01$ |
| LLaMA | Baseline | $10.75 \pm 0.32$ | $10.55 \pm 0.07$ | $10.72 \pm 0.10$ | $11.06 \pm 0.25$ |
| | RaLMSpec | $11.47 \pm 0.17$ | $11.02 \pm 0.31$ | $11.10 \pm 0.27$ | $10.79 \pm 0.20$ |
| | RaLMSpec+P(20) | $11.27 \pm 0.14$ | $11.04 \pm 0.22$ | $10.69 \pm 0.15$ | $10.66 \pm 0.23$ |
| | RaLMSpec+P(256) | $12.83 \pm 0.21$ | $13.35 \pm 0.81$ | $12.48 \pm 0.22$ | $12.60 \pm 0.36$ |
| | RaLMSpec+S | $10.38 \pm 0.28$ | $10.19 \pm 0.19$ | $9.95 \pm 0.04$ | $10.18 \pm 0.08$ |
| | RaLMSpec+A | $10.88 \pm 0.26$ | $10.66 \pm 0.12$ | $10.56 \pm 0.20$ | $10.16 \pm 0.18$ |
| | RaLMSpec+P(20)SA | $\mathbf{8.28 \pm 0.18}$ | $\mathbf{8.18 \pm 0.10}$ | $\mathbf{8.26 \pm 0.14}$ | $\mathbf{8.09 \pm 0.18}$ |
| | RaLMSpec+P(256)SA | $9.46 \pm 0.17$ | $10.42 \pm 0.74$ | $9.64 \pm 0.08$ | $9.36 \pm 0.16$ |

