# OpenReview forum: "Accelerating Retrieval-augmented Language Model Serving with Speculation"
_ICLR.cc/2024/Conference — Submitted to ICLR 2024_

### Official Review · Reviewer_17wi · 2023-10-29

**Soundness:** 3 good
**Presentation:** 3 good
**Contribution:** 3 good
**Rating:** 6
**Confidence:** 3

**Summary:**

This paper aims to accelerate retrieval-augmented language models (RaLM) by incorporating speculation over iterative RaLM. The proposed framework, RaLMSpec, leverages the temporal locality of the retrieved documents and uses a caching-based speculative retrieval mechanism with bached verification to reduce the latency in the retrieval phase. Based on this, three techniques are proposed to further speed up the process, i.e., top-k cache update, asynchronous verification, and optimal speculation stride scheduler. Experiments are conducted on various datasets, language models, and retrievers and RaLMSpec is proven to be effective.

**Strengths:**

- This paper targets an interesting and promising direction in the community, which is to make the inference faster so that the models could be more deployable in real-world scenarios.
- This is the first work to employ the idea of speculation in retrieval-augmented language models.
- The proposed framework is sound and the paper is overall well-written and organized. The figures demonstrated the method vividly. Almost everything is clear to me.
- The proposed methods are experimented with comprehensive settings, and the results are significant when using dense retrievers.

**Weaknesses:**

- The proposed framework, with three additional tricks stacked, is still not promising in all settings. For example, it does not bring much speedup with approximate dense retrievers on GPT and Llama-2.
- The related work section needs reorganization. The current version simply decomposes the framework into related parts and introduces the works one by one. It's hard to grasp the major contribution and position of this work correctly at first glance.

**Questions:**

1. How will the *speculation stride s* evolve during the process? Is there any experiments/analysis on how the initialization of *s* affect the performance?

2.  Any intuitions on why RaLMSpec brings more speedup on OPT2 than the other two models?

---

> ### Author Response · Authors · 2023-11-15
> **Respond to reviewer 17wi**
>
> We thank the reviewer for their review and address their main concerns below.
>
> `Q1. Speculation Stride`
>
> When we don’t enable the optimal speculation stride scheduler, the stride of speculation is prefixed to 3. However, if the optimal speculation stride scheduler (OS$^3$) is enabled, the speculation stride will be initialized to 1 and dynamically adapted according to the theoretical optimal value derived from the cost model. Speculation stride is indeed a crucial hyperparameter that controls the tradeoff between speculation gain and overhead. We include an ablation Table 5 and a more fine-grained analysis in Appendix A.6 to show how different speculation stride affects the performance. We also show the table below. The values are the average serving latency per request.
>
> | Retriever |   s=2  |     s=4    |   s=8  |   OS$^3$   |
> |:---------:|:------:|:----------:|:------:|:----------:|
> |    EDR    | 92.17s | **81.06s** | 81.90s |   85.19s   |
> |    ADR    |  9.86s |   14.93s   | 25.88s |  **8.14s** |
> |     SR    | 10.65s |   12.48s   | 16.66s | **10.38s** |
>
> We can observe that larger a stride is better for EDR while smaller a stride is better for ADR/SR. OS$^3$ can capture this dynamism and achieve near-optimal performance. The reason why OS$^3$ performs slightly worse than $s=4, 8$ in the case of EDR is because a warm-up phase is required for OS$^3$ to start its adaptation, i.e., we initialize s=1 when we enable OS$^3$. Thus, the warm-up phase can introduce non-optimal strides and, thus, slightly worse performance.
>
> Besides, we have also added additional experiments on our approach against KNN-LM [1] which is another important RaLM workload. The analysis and results are included in Appendix A.3 of the revised paper. The KNN-LM results also include the ablation study on the speculation stride and our detailed analysis. As a retrieval-intensive workload, our approach can achieve up to 7.59x speed up when the optimal speculation stride scheduler is used.
>
> `Q2. OPT2 performance`
>
> As the different models can generate contents differently and thus have different retrieval patterns, the main reason is that the OPT2 model tends to generate contents that have a retrieval pattern that is easier to speculate (e.g., more documents have been repeatedly retrieved), which results in a higher speculation accuracy and thus higher speed-up ratio.
>
> `Q3. ADR performance`
>
> As our approach targets accelerating the retrieval part of the RAG workloads, and the reported speed-up is the end-to-end speed-up ratio, the speed-up ratio is intrinsically upper-bounded by the retrieval ratio with respect to the overall latency. Thus, in the case of the approximate dense retriever, the retrieval latency is much faster than the language model generation latency, which results in a limited acceleration space for our approach, as shown in Figure 1. However, we want to demonstrate that our method can consistently achieve speed-ups even in cases where the acceleration space is limited.
>
> `Q4. Related Work`
>
> We thank the reviewer for pointing out the drawbacks of our related work presentation, and we have modified the related work section accordingly in the revised paper by reorganizing paragraphs and pruning out unnecessary descriptions to focus more on the position and contribution of our work within the literature.
>
> References:
>
> [1] Khandelwal, Urvashi, et al. "Generalization through memorization: Nearest neighbor language models." International Conference on Learning Representations. 2020.

---

### Official Review · Reviewer_ee61 · 2023-10-29

**Soundness:** 3 good
**Presentation:** 3 good
**Contribution:** 3 good
**Rating:** 5
**Confidence:** 4

**Summary:**

This author proposes a new framework called RaLMSpec that accelerates retrieval-augmented language model serving with speculation. RaLMSpec provides generic speed-up over iterative RaLM while preserving the same model outputs through speculative retrieval and batched verification.

The author empirically evaluated RaLMSpec on three language models on four downstream QA datasets, and the results show that RaLMSpec achieves a speed-up ratio of up to 2.39× compared to the iterative RaLM approach.

Key components include:

1. Caching-based speculative retrieval mechanism and prefetching: RaLMSpec leverages the temporal locality of retrieved documents to enable a request-level local cache for speculative retrieval.  RaLMSpec prefetches the cache to reduce the latency of the first retrieval.

2. Batched and asynchronous verification: RaLMSpec uses a batched verification step to guarantee correctness while reducing the retrieval overhead. This is done asynchronously to overlap the verification with the retrieval and further boost the speculation performance.

3. Optimal speculation stride scheduling: RaLMSpec schedules the speculation stride optimally to balance the trade-off between speed and accuracy.

**Strengths:**

1. Empirical results are impressive: RaLMSpec achieved better performance in terms of speed and accuracy. In the meantime, RaLMSpec achieved comparable or better accuracy than the baseline approach while providing a significant speed-up.

2. The presentation and the technique proposed are clear: batching, prefetching, asynchronicity, and scheduling. While none of this is new standalone, the authors combine them clearly and cleverly to boost the overall performance of augmented retrieval.

**Weaknesses:**

1. The ablation study does not shed light on which component contributes the most to the performance gain. It would be nice to have a breakdown on how much batching, prefetching, asynchronicity, and scheduling contributed to the overall performance improvement to give more insights about the characteristics of the workload.

**Questions:**

My main concern is regarding the ablation study listed above. I would be happy to raise my rating if the author can provide further analysis on how each component contribute to the overall performance gain and share more insights about the characteristics of the workload.

---

> ### Author Response · Authors · 2023-11-15
> **Respond to reviewer ee61**
>
> We thank the reviewer for their review and address their main concerns below.
>
> `Q1. Ablation Study`
>
> We thank the reviewer for acknowledging our contribution. Indeed, the proposed techniques (cache-based speculative retrieval, prefetching, optimal speculation stride scheduler, and asynchronous verification) are not orthogonal to each other. Thus, it is not as trivial as it seems to directly decompose the speed-up ratio to different techniques. However, we conduct experiments for each technique individually as well as all together, as shown in Table 1, where P stands for prefetching, S stands for optimal speculation stride scheduler and A stands for asynchronous verification. If we ignore the PSA results (the one that enabled all techniques), we can see from Table 1 that the optimal speculation stride scheduler contributes most to the speed-up in most cases compared with prefetching and asynchronous verification. This is due to the fact that the speculation stride is a crucial hyperparameter that controls the tradeoff between speculation gain and overhead, and the proposed OS^3 technique can adaptively find a theoretical optimal value. Besides, we have also included an additional Table 4, Figure 7, and more fine-grained analysis in Appendix A.5 of the revised paper to further show the contribution of each component. We have also included the additional ablation table below. The results are averaged serving latency per request.
>
> | Retriever | Baseline |    P   |    S   |    A   |   PS   |     SA    |     PA     |     PSA    |
> |:---------:|:--------:|:------:|:------:|:------:|:------:|:---------:|:----------:|:----------:|
> |    EDR    |  144.39s | 82.23s | 85.19s | 90.49s | 81.64s |   85.13s  | **81.60s** | **79.06s** |
> |    ADR    |   8.06s  | 14.25s |  8.14s | 13.90s |  8.17s | **7.83s** |   12.84s   |  **7.89s** |
> |     SR    |  10.75s  | 11.27s | 10.38s | 10.88s | 10.21s | **8.26s** |   10.61s   |  **8.28s** |
>
> We observe that prefetching works pretty well when an exact dense or sparse retriever is used because the overhead of doing prefetching is relatively small compared to when an approximate dense retriever is used. For workloads with a retrieval latency similar to the language model generation latency, asynchronous verification can bring up the most significant speed when optimal speculation stride scheduler is enabled but not on itself. To sum up, the optimal speculation stride scheduler is the most important component while the prefetching and asynchronous verification can be beneficial depending on which retriever or language model is being used.
>
> `Additional Experiments`
>
> To provide more information to the reviewer, we have also added additional experiments on KNN-LM [1] with our approach in Appendix A.3 of the revised paper, which further verifies that our method can achieve latency savings consistently in different scenarios. As a retrieval-intensive workload, our approach can achieve up to 7.59x speed up when the optimal speculation stride scheduler is used.
>
> References:
>
> [1] Khandelwal, Urvashi, et al. "Generalization through memorization: Nearest neighbor language models." International Conference on Learning Representations. 2020.

---

### Official Review · Reviewer_RYWk · 2023-10-29

**Soundness:** 3 good
**Presentation:** 3 good
**Contribution:** 2 fair
**Rating:** 3
**Confidence:** 3

**Summary:**

This paper proposes a framekwork called RaLMSpec for accelerating iterative RaLM serving. It is inspired by speculative execution and speculative decoding and reduces the number of calls to retrieve from external corpora so that decreases its serving cost. The main idea sounds reaonable and the experiment results show moderate speedup.

**Strengths:**

The idea of speculative retrieval sounds interesting and may reduce the number of calls to the large external corpora. It can potentially save cost of serving an iterative RAG system.

**Weaknesses:**

I am primarily concerned about the significance of this work.

Firstly, I am unsure about the practicality of iterative RAG. From my understanding, current AI-powered search engines like Bing Chat do not operate iteratively. Instead, they perform retrieval all at once and then apply RAG. These systems typically have a cache system that stores the results of common queries, thereby saving the cost of performing retrieval for each query. The only difference between this paper and these common practices is the iterative RAG setting. However, I am still concerned about this iterative setting, as I am uncertain whether the benefits are significant enough, especially for LLMs.

Secondly, as far as I understand, the main cost of the RAG model lies in the generation aspect. For mature search engines, various optimizations of the retrieval part are already well-established, fast, and efficient. However, generation has always been the more costly part. Therefore, I am unsure whether the speedup improvement reported by the authors refers to the savings in the retrieval part or the end-to-end savings. If it is the latter, I would like to know how much improvement this technology could bring when applied to a mature search engine.

Some minor issues:

- A comparison between iterative RAG and conventional RAG should be presented to demonstrate the value of iterative RAG, as it may introduce more latency.

- The LLMs used in the experiments are small. This makes the retrieval speed improvement appear greater in the end-to-end latency improvement. However, when the LLM becomes larger, the end-to-end speed improvement will become very marginal.

- There are previous related studies (e.g., [1]) that explore the idea of speculation and cache in the RAG setting. The authors should include these as related work, while they focus more on the generation speedup.

References:

[1] Yang et al.: Inference with Reference: Lossless Acceleration of Large Language Models. https://arxiv.org/abs/2304.04487

**Questions:**

See the weakness section

---

> ### Author Response · Authors · 2023-11-15
> **Respond to reviewer RYWk Part (1/2)**
>
> We thank the reviewer for their review and address their main concerns below.
>
> `Q1. The practicality of iterative RAG`
>
> We agree with the reviewer that one-shot RAG, where retrieval is conducted once at the start, is a major trend for now in most RAG applications (e.g., LlamaIndex, LangChain). However, the generation quality of such applications largely relies on a powerful language model like GPT-3.5. We note that even in such cases, iterative retrieval can outperform one-shot retrieval (e.g., FLARE [2], IRCoT [4]), especially for longer conversations. A more reasonable scenario for iterative retrieval is when you don’t have access to an LLM as powerful as GPT-3.5 but can use smaller surrogates like GPT-2, OPT-1.3B, or LLaMA-2-7B. In such scenarios, single-shot retrieval performs much worse than iterative retrieval, as shown in IC-RaLM [3] and IRCoT [4]. Additionally, KNN-LM [1] is another retrieval-intensive workload that has been shown to improve the performance of the base language model with retrieval augmentation. To demonstrate our approach also works for KNN-LM, we provide additional experiments in Appendix A.3 of our revised paper to verify the effectiveness of RaLMSpec. More specifically, our approach can achieve a speed-up ratio of up to 7.59x compared with the baseline implementation as shown in Figure 6.
>
> `Q2. The difference with the Caching mechanism in Bing-Chat-like systems`
>
> The local cache in our system design is used for speculative retrieval instead of query matching. In the scenario where different queries might retrieve the same entry in the cache, Bing-Chat-like systems will have a cache miss due to query mismatch, but our design will still return a retrieved entry in the cache. This design is two-sided. On the one hand, it can benefit from a correct speculative retrieval. On the other hand, we need to add an additional verification mechanism to correct false speculations. The interaction between speculation and verification is crucial (e.g., how many speculation steps will be performed before a verification step, which we call the speculation stride). To this end, we further propose the optimal speculation stride scheduler to automatically adjust to the theoretical optimal strides. To sum up, though our system design also leverages the caching mechanism, the cache is used differently (speculative retrieval rather than common-entry matching); thus, additional efforts are required to guarantee correctness.
>
> `Q3. Mature search engines`
>
> The reported speed-up ratios are all against end-to-end latencies. Our approach is intrinsically targeting reducing the retrieval latency, not the model generation latency. Thus, if the proportion of the retrieval latency is relatively small compared to the overall latency, the speed-up ratio can be limited, as shown in Figure 4. However, we do believe that there are certain trade-offs between retrieval efficiency and model generation efficiency in different setups, where some prefer a slower but more accurate retriever while others might prefer efficiency over accuracy. However, we want to note that our approach can automatically adjust to the optimal configurations in all different scenarios and achieve speed-ups accordingly, which is enabled by prefetching, the optimal speculation stride scheduler, and the asynchronous verification techniques we proposed in the paper. This can be further verified by our original and updated empirical results. Regarding how well the proposed method will work with mature search engines, it depends on the proportion of retrieval latency over the total serving latency. If the retrieval operation is optimized by approximation and parallelization, the acceleration space of our approach can be limited. However, the proposed algorithm can achieve a more significant speed-up for slow retrieval scenarios due to hardware/software limitations.
>
> `Q4. Comparison between one-shot and iterative RAG`
>
> We thank the reviewer for pointing out the need for this comparison. As our paper mainly focuses on accelerating iterative RAG while preserving the same model output, we refer the reviewer to [1, 2, 3, 4] for more detailed motivation and experimental results of iterative RAG. For one, Table 1 in [1] has shown that KNN-LM can consistently outperform the base language model with retrieval. Figure 5 in [3] also shows that with a more frequent retrieval step, the generation quality (measured by perplexity) of the RAG model can be improved monotonically.

---

> > ### Author Response · Authors · 2023-11-15
> > **Respond to reviewer RYWk Part (2/2)**
> >
> > `Q5. Larger LLMs`
> >
> > We have included additional experiments on LLaMA-2-13B and summarized the results below as well as in Appendix A.4 of the revised paper. Since most iterative RAG methods work with a smaller LM and we currently don’t have enough resources and time to conduct experiments with models larger than 13B, hopefully, our added experiments can partially address the reviewer’s concern.
> >
> > | Retriever | Wiki QA | Web Questions | Natural Questions | Trivia QA |
> > |:---------:|:-------:|:-------------:|:-----------------:|:---------:|
> > |    EDR    |  1.70x  |     1.85x     |       1.73x       |   1.78x   |
> > |    ADR    |  1.03x  |     1.04x     |       1.02x       |   1.03x   |
> > |     SR    |  1.18x  |     1.21x     |       1.22x       |   1.26x   |
> >
> > `Q6. Related Works`
> >
> > We thank the reviewer for pointing out the missing related works. We have updated the related work section accordingly in the revised paper.
> >
> > References:
> >
> > [1] Khandelwal, Urvashi, et al. "Generalization through memorization: Nearest neighbor language models." International Conference on Learning Representations. 2020.
> >
> > [2] Jiang, Zhengbao, et al. "Active retrieval augmented generation." To appear in the Conference on Empirical Methods in Natural Language Processing. 2023.
> >
> > [3] Ram, Ori, et al. "In-context retrieval-augmented language models." To appear in Transactions of the Association for Computational Linguistics. 2023.
> >
> > [4] Trivedi, Harsh, et al. "Interleaving retrieval with chain-of-thought reasoning for knowledge-intensive multi-step questions." Proceedings of the 61st Annual Meeting of the Association for Computational Linguistics. 2023.

---

### Official Review · Reviewer_pUX8 · 2023-11-07

**Soundness:** 3 good
**Presentation:** 3 good
**Contribution:** 2 fair
**Rating:** 3
**Confidence:** 3

**Summary:**

This paper presents a collection of methods aimed to accelerate the retrieval-augmented language model. In particular, RaLMSpec is proposed, which employs speculation-inspired framework that provides generic speed-up over iterative RaLM while preserving the same model outputs through speculative retrieval and batched verification. Additionally, prefetching, optimal speculation stride scheduler and synchronous verification are further proposed to improve the performance. Experimental evaluation show a speed-up ratio of around 2x compared to the baseline method.

**Strengths:**

1. It presents several techniques to speedup the retrieval-augmented language model serving.
2. It achieves speedup over the baseline approach and seems satisfactory.

**Weaknesses:**

1. The overall contribution of the paper is not quite enough to present in ICLR.
2. The novelty of the paper is limitted.

**Questions:**

1. The the speedup varies from different settings (e.g. hardware, data, model, etc)?
2. why the speed up for different retriever differ so significantly? It is better the show breakdown the the major time-consuming parts and perform insightful analysis.

---

> ### Author Response · Authors · 2023-11-15
> **Respond to reviewer pUX8**
>
> We thank the reviewer for their review and address their main concerns below.
>
> `Q1. Contribution and Novelty`
>
> Though the idea of speculative execution is not new, as far as we know, we are the first work that incorporates the speculation idea into accelerating retrieval augmented language model serving. More specifically, we design a local retrieval cache to enable speculative retrieval that is novel and unseen before. In addition to the design for a local retrieval cache, we introduce prefetching, optimal speculation stride scheduler, and asynchronous verification to boost the performance further. While prefetching and asynchronous verification are relatively trivial to think of, the optimal speculation stride scheduler is a cost model-based dynamic speculation scheduler that can automatically adjust the speculation stride to the optimal setup in various scenarios, which is novel and effective.
>
> Besides technical contributions, we also evaluate our approach against three different types of retrievers and three different sizes of LMs (GPT, OPT, and LLaMA) across four QA datasets. The overall results and ablation study demonstrate that our method with all components enabled (prefetching+OS3+AsynchVerify) can consistently achieve the best speed-up ratio. We further include additional experimental results for the KNN-LM[1] workload in Appendix A.3 as well as the results for LLaMA-2-13B in Appendix A.4. For KNN-LM, our approach can achieve up to 7.59x speed-up ratio. For LLaMA-2-13B, our approach can still achieve a speed-up from 1.02x to 1.85x.
>
> With the aforementioned technical and empirical contributions, we hope the reviewer can reconsider the novelty and contribution of our work.
>
> `Q2. Speed-up Variations`
>
> The speed-up ratio varies because different retrievers, language models, or datasets are used. More specifically, as our approach is intrinsically optimizing the retrieval latency, but we are reporting the end-to-end latency speed-up ratios, different retrievers and different language models might contribute differently (latency-wise) in the end-to-end latency. This can be demonstrated in Figure 4, where each latency bar constitutes two sub-bars, one for retrieval latency and one for language model generation latency.  When an approximate retriever is used, the retrieval latency will not dominate the serving latency, thus the end-to-end speed-up ratio is limited. When a dense retriever is used, the retrieval overhead is much higher, therefore, a higher speed-up ratio can be achieved accordingly. Figure 4 and the analysis paragraph in section 5.2 present the figure and written explanation. In order to make this more clear, we have updated the analysis further in the revised paper.
>
> References:
>
> [1] Khandelwal, Urvashi, et al. "Generalization through memorization: Nearest neighbor language models." International Conference on Learning Representations. 2020.

---

### Author Response · Authors · 2023-11-15
**General respond**

We thank the reviewers for all their constructive comments. Based on the comments (please refer to our individual replies for details), we have prepared a revised submission, where we focus on clarifying the most important confusions we identified and providing additional results as much as possible. All revised texts are in blue. Most of the additional experimental results are included in Appendix A.3-6.

Summary:
1. Reorganized and added missing citations to the related work section to connect our work with the RaLM literature better.
2. Added more descriptions in the evaluation section.
3. Added the KNN-LM [1] experiments to further support the effectiveness of our approach in Appendix A.3 of the revised paper.
4. Added the LLaMA-2-13B model experiments to further support the effectiveness of our approach against larger language models in Appendix A.4 of the revised paper.
5. Added additional ablation studies on the contribution of different components (prefetching, optimal speculation stride scheduler, and asynchronous verification) in Appendix A.5 of the revised paper.
6. Added additional ablation studies on the speculation stride in Appendix A.6 of the revised paper.

Reference:

[1] Khandelwal, Urvashi, et al. "Generalization through memorization: Nearest neighbor language models." International Conference on Learning Representations. 2020.

---

### Author Response · Authors · 2023-11-21
**Rebuttal discussion**

We thank the reviewer for their time to provide insightful comments. We thus responded to the comments and revised the paper with additional experiments and a better presentation (writing) accordingly. We understand the reviewers can have other reviewing or submission tasks but kindly hope the reviewers can provide any feedback or additional concern if any, for which we can further address.

---

### Meta-Review · Area_Chair_teiw · 2023-12-15

**Metareview:**

I believe that one of the reviewers summarized well reseasons that this paper is not ready for publication at ICLR:

I am primarily concerned about the significance of this work.

Firstly, I am unsure about the practicality of iterative RAG. From my understanding, current AI-powered search engines like Bing Chat do not operate iteratively. Instead, they perform retrieval all at once and then apply RAG. These systems typically have a cache system that stores the results of common queries, thereby saving the cost of performing retrieval for each query. The only difference between this paper and these common practices is the iterative RAG setting. However, I am still concerned about this iterative setting, as I am uncertain whether the benefits are significant enough, especially for LLMs.

Secondly, as far as I understand, the main cost of the RAG model lies in the generation aspect. For mature search engines, various optimizations of the retrieval part are already well-established, fast, and efficient. However, generation has always been the more costly part. Therefore, I am unsure whether the speedup improvement reported by the authors refers to the savings in the retrieval part or the end-to-end savings. If it is the latter, I would like to know how much improvement this technology could bring when applied to a mature search engine.

**Justification For Why Not Higher Score:**

See the main comments.

**Justification For Why Not Lower Score:**

This is the lowest score.

---

### Decision · Program_Chairs · 2024-01-16

Reject